# A Preliminary Study of o1 in Medicine: Are We Closer to an AI Doctor 🤖?

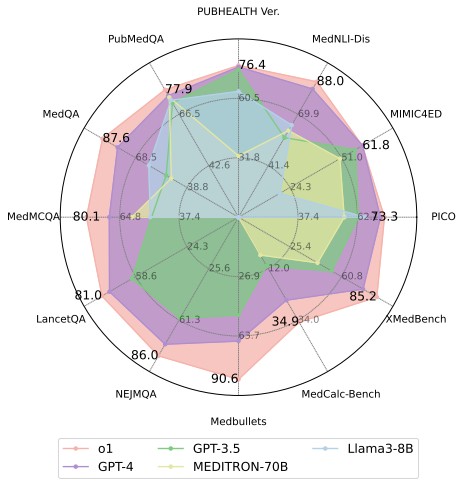

Figure 1: Overall results of o1 and other 4 strong LLMs. We show performance on 12 medical datasets spanning diverse domains. o1 demonstrates a clear performance advantage over close- and open-source models.

Figure 2: Average accuracy of o1 and other 4 strong LLMs. o1 achieves the highest average accuracy of 74.3% across 19 medical datasets.

## ABSTRACT

Large language models (LLMs) have exhibited remarkable capabilities across various domains and tasks, pushing the boundaries of our knowledge in learning and cognition. The latest model, OpenAI's o1, stands out as the first LLM with an internalized chain-of-thought technique using reinforcement learning strategies. While it has demonstrated surprisingly strong capabilities on various general language tasks, its performance in specialized fields such as medicine remains unknown. To this end, this report provides a preliminary exploration of o1 on different medical scenarios, comprehensively examining 3 key aspects: *understanding*, *reasoning*, and *multilinguality*. Specifically, our evaluation encompasses 6 tasks using data from 37 medical datasets, including two newly constructed and more challenging question-answering (QA) tasks based on professional medical quizzes from the New England Journal of Medicine and The Lancet. These datasets offer greater clinical relevance compared to standard medical QA benchmarks such as MedQA, translating more effectively into real-world clinical utility. Our analysis of o1 suggests that the enhanced reasoning ability of LLMs may (significantly) benefit their capability to understand various medical instructions and reason through complex clinical scenarios. Notably, o1 surpasses the previous GPT-4 in accuracy by an average of 6.2% and 6.6% across 19 datasets and two newly created complex QA scenarios. But meanwhile, we also identify several weaknesses in both the model capability and the existing evaluation protocols, including hallucination, inconsistent multilingual ability, and discrepant metrics for evaluation. We will release our raw data and model outputs for future research.

# 1 INTRODUCTION

Intelligence, a complex and elusive concept, has puzzled psychologists, philosophers, and computer scientists for years (Bubeck et al., 2023). While there is no single agreed-upon definition of intelligence, it is widely accepted that it spans a broad range of cognitive skills, rather than being confined to a specific task (McCarthy et al., 1955). Creating artificial systems with such general intelligence has been a long-standing and ambitious goal of AI research. The most exciting progresses in AI are achieved by language models in these years, from the initial start of ChatGPT to its evolution and other open-source projects (Touvron et al., 2023a;b; Jiang et al., 2023; Bai et al., 2023; Peng et al., 2024).

Early LLM pioneers set out goals to understand and interact with human by exploring generalizable reasoning mechanisms and building knowledge bases with vast amounts of commonsense information. With parameters and data volume in place, the question of how to effectively prompt the model from the user end and train it from the developer end has become a trending topic of exploration (Wei et al., 2022; Ouyang et al., 2022). On the user side, varying prompting techniques can significantly impact model performance. Chain-of-thought (CoT) prompting (Wei et al., 2022; Dong et al., 2022; Saunders et al., 2022), one of the most popular strategies, leverages the model's internal reasoning patterns to enhance its ability to solve complex tasks. OpenAI capitalized on this by embedding the CoT process into model training, integrating reinforcement learning, and finally introduced the `o1` model (OpenAI, 2024). While the `o1` model demonstrates strong performance in general domains, its effectiveness in specialized fields like medicine—where domain-specific training may be lacking—remains uncertain. Moreover, current benchmarks for LLMs in the medical domain often evaluate models only on a limited set of factors, often focusing on isolated aspects such as knowledge and reasoning (Nori et al., 2023b; Liévin et al., 2024), safety (Han et al., 2024), or multilinguality (Wang et al., 2024). These factors make a comprehensive assessment of LLMs' capabilities—especially for advanced models like `o1`—in medical challenging tasks (Figure 1).

This paper aims to provide an initiative to close this gap, focusing on `o1`. We identify three fundamental aspects of LLMs in medicine: *understanding*, *reasoning*, and *multilinguality*. To evaluate these capabilities, we assembled 35 existing medical datasets and developed two novel, challenging QA datasets that include instructions and expected outputs, ensuring comprehensive assessment. With evaluation on this extensive suite, our key findings include:

- `o1` demonstrates improved transfer of clinical understanding and reasoning abilities, validating its competence in real-world diagnostic scenarios compared with both close- and open-source models as presented in Figure 1 and Figure 2;
- No single model excels across all tasks on our medical leaderboard, though `o1` comes close to dominating most evaluations;
- `o1` still suffers from the long-standing issue of hallucination and complex multilingual medical cases;
- Inconsistencies in metrics for medical NLP can significantly affect models' standings, which calls for a re-evaluation of reliable metrics for future LLMs;
- CoT prompting can further enhance `o1` in medicine, despite its training having already integrated CoT data.

In addition to these findings, we also elevate the discussion section as an initial attempt to address the issues identified during our benchmarking in Section Section 5. Particularly, we highlight the potential negative effects of `o1`, emphasize the urgent need for consistent and unified evaluation metrics for future LLMs, and advocate for improved instruction templates that can be applied to models with embedded prompting strategies.

# 2 RELATED WORKS

**Large Language Models with Enhanced Reasoning Ability.** Large Language models (LLMs) based on next token prediction pre-training (Touvron et al., 2023a;b; Achiam et al., 2023) have demonstrated promising capabilities on various language undersanding tasks. Instruction fine-tuning further improved the abilites of these LLMs for following user instructions. However, recent studies

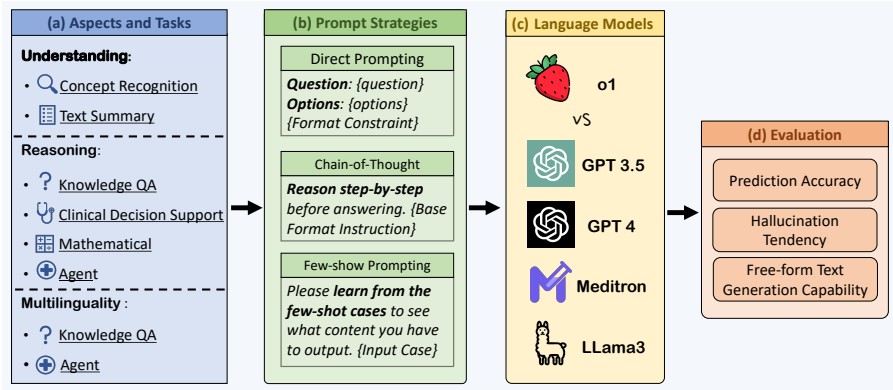

Figure 3: Our evaluation pipeline has different (a) *aspects* with various (b) *prompting strategies* using the latest (c) *language models*. We leverage a comprehensive set of (d) *evaluations* to present a holistic view of model progress in the medical domain.

suggest that LLMs struggle with complex tasks involving logical reasoning. To address this issue, some researches propose to instruct LLMs to mimic human thinking processes by producing a chain-of-thought (CoT) (Feng et al., 2024; Wei et al., 2022) before generating a final answer. Reinforcement learning from human feedback (Ouyang et al., 2022) has also been employed to enhance reasoning while make sure the models align with human values (Tu et al., 2023b;a). Recently, OpenAI introduced o1, which was trained on a vast amount of CoT data, further enhancing the capability of LLMs in solving scientific problems. In this paper, we aim to investigate whether enhanced abilities of o1 effectively transfer to the clinical medical domain.

**Medical Large Language Models.** Benefiting from the generalization capabilities of LLMs, general-purpose models such as GPT-4 have demonstrated impressive performance on challenging medical problems (Nori et al., 2023a; Wu et al., 2024b). Some researchers have attempted to further equip LLMs with biomedical knowledge by fine-tuning them using domain-specific corpora (Chen et al., 2023; Wang et al., 2023; Wu et al., 2024a; Li et al., 2023). However, for clinical applications, LLMs are not only required to understand medical domain-specific knowledge but also to produce reliable responses by performing logical reasoning. In this paper, we aim to explore the potential of o1 as a clinical viable model. Our experimental findings reveal that with enhanced *understanding*, *reasoning*, and *multilinguality* medical capabilities, o1 makes a step closer to reliable clinical AI-system.

## 3 EVALUATION PIPELINE

### 3.1 OVERALL TAXONOMY OF EVALUATIONS

First, we present the taxonomy of our evaluation, along with an overview of the evaluation pipeline as shown in Figure 3. Firstly, we specify three aspects of the model capabilities, namely *understanding*, *reasoning*, and *multilinguality*, that correspond to the real-world needs of clinical physicians. To ensure a comprehensive evaluation, we collect a diverse range of medical tasks and datasets that fall under these three aspects. Moreover, we explore three prompting strategies in our pipeline, including (1) direct prompting, which instructs LLMs to solve specific problems directly, (2) chain-of-thought, which requires models to think step-by-step before generating the final answer, (3) few-shot prompting, which providing models with several examples to learn the input-output mapping on the fly. Lastly, appropriate metrics are utilized to measure the discrepancy between generated responses and ground-truth answers. Details about metrics utilized in each dataset are provided in Table 1.

---

[1] https://www.thelancet.com/
[2] https://www.nejm.org/

Table 1: Six tasks across three fundamental aspects employed in our evaluation suite. Asterisks (*) denotes the newly constructed datasets from public sources.

| Aspect | Task | Dataset | Description | Metrics |
|---|---|---|---|---|
| Understanding | Concept Recognition | BC5-disease (Li et al., 2016) | Entity extraction for disease. | F1-score |
| | | BC5Chem (Li et al., 2016) | Entity extraction for chemical. | |
| | | BC4Chem (Savery et al., 2020) | Entity extraction for chemical names from PubMed article abstracts. | |
| | | Species800 (Pafilis et al., 2013) | Extraction of organism names from PubMed article abstracts. | |
| | | HoC (Baker et al., 2016) | Classification of the hallmarks of cancer given biomedical article abstracts. | |
| | | HumanDiseaseOntology (Schriml et al., 2019) | Disease ontology-based entity extraction. | BLEU, ROUGE, AlignScore, Mauve |
| | | BioLORD (Remy et al., 2024) | Elaboration of biomedical concepts. | |
| | | PMC-Patient (Zhao et al., 2023) | Patient-related entity (gender and age for example) extraction from PubMed Central articles. | Accuracy |
| | | PICO-Participant (Nye et al., 2018) | Information extraction of outcome, intervention, and participant from article abstracts. | |
| | | PICO-Intervention (Nye et al., 2018) | | |
| | | PICO-Outcome (Nye et al., 2018) | | |
| | | ADE Corpus (Gurulingappa et al., 2012) | Drug dose extraction given the drug information. | |
| | Text Summary | MIMIC-IV-Ultrasound (Johnson et al., 2023) | Summarization of patient reports from emergency departments. | BLEU, ROUGE, AlignScore, Mauve |
| | | MIMIC-IV-CT (Wallace et al., 2021) | | |
| | | RCT-Text (Wallace et al., 2021) | Summarization of medical evidence from clinical studies in literature reviews. | |
| | | MedQSum (Lee et al., 2021) | Summarization of patient notes, reports, and health records. | |
| Reasoning | Knowledge QA | PubMedQA (Jin et al., 2019) | QA data built on PubMed abstracts. | Accuracy |
| | | MedQA (Jin et al., 2021) | QA data for medical knowledge assessment. | |
| | | MedMCQA (Pal et al., 2022) | QA data from AIIMS & NEET PG entrance exams. | |
| | | LancetQA [1] | QA data crawled from Lancet picture quiz gallery. | |
| | | NEJMQA [2] | QA and diagnostic challenge requests from NEJM quiz. | |
| | | Medbullets (Chen et al., 2024) | QA data from Medbullets online medical study platform. | |
| | Clinical Decision Support | DDXPlus (Fansi Tchango et al., 2022) | Diagnostic decision making of synthesized patient data. | Accuracy |
| | | SEER (Dubey et al., 2023) | Treatment planning for breast cancer cases. | |
| | | MIMIC4ED-Hospitalization (Xie et al., 2022) | Prediction of clinical outcomes in emergency medicine from MIMIC-IV-ED. | |
| | | MIMIC4ED-72h ED Revisit (Xie et al., 2022) | | |
| | | MIMIC4ED-Critical Triage (Xie et al., 2022) | | |
| | | MedNLI-Dis. (Romanov & Shivade, 2018) | Discriminative entailment task for clinical hypotheses. | |
| | | PUBHEALTH Ver. (Kotonya & Toni, 2020) | Verification of health-related information from the public. | |
| | | EBMS (Mollá & Santiago-Martinez, 2011) | Justification verification using the EBMS corpus. | BLEU, ROUGE, AlignScore, Mauve |
| | | PUBHEALTH Exp. (Kotonya & Toni, 2020) | Explanation of health-related information from the public. | |
| | | ChatDoctor (Li et al., 2023) | Patient-doctor dialogues from online medical consultations. | |
| | | MedNLI-Gen. (Romanov & Shivade, 2018) | Generative entailment task for clinical hypotheses. | |
| | Agent | AI Hospital (Fan et al., 2024) | Multi-agent task simulating dynamic medical interactions in Chinese. | Accuracy |
| | | AgentClinic (Schmidgall et al., 2024) | Agent benchmark in simulated clinical environments from MedQA and NEJMQA scenarios. | |
| | Medical Calculation | MedCalc-Bench (Khandekar et al., 2024) | Medicine dose level calculation from ADE corpus. | Accuracy |
| Multi-linguality | Knowledge QA | XMedBench (Wang et al., 2024) | Multilingual benchmark for medical understanding and interaction. | Accuracy |
| | Agent | AI Hospital (Fan et al., 2024) | Multi-agent task simulating dynamic medical interactions in Chinese. | Accuracy |

## 3.2 ASPECTS AND TASKS

In Table 1, our evaluation efforts are structured into three main parts: aspect, task, and dataset. Specifically, a **dataset** refers to the data itself along with the metrics used in the current context. We utilize 35 existing datasets and create 2 additional challenging datasets for evaluation. A **task** is a collection of multiple datasets that share a common goal or evaluate similar capabilities within the model. We categorize all 37 datasets into 6 tasks for clearer evaluation and analysis. An **aspect**

describes a specific capability or property to understand how well the model performs in a particular area. In our evaluation pipeline, we focus on three key aspects.

Formally, we illustrate these three evaluation aspects with their corresponding tasks as follows:

- **Understanding** refers to the model's ability to utilize its internal medical knowledge to comprehend medical concepts. For example, in concept recognition task, the model is required to extract or elaborate medical concepts from article (Savery et al., 2020; Pafilis et al., 2013; Nye et al., 2018) or diagnosis report (Zhao et al., 2023). And in text summarization, the model need to understand concepts in complex texts to generate a concise summary (Lee et al., 2021; Wallace et al., 2021; Johnson et al., 2019; 2023).

- **Reasoning** is the ability to conduct multiple steps of logical thinking to arrive at the conclusion. In question answering tasks, the model is prompted to select correct option from multi-choices based on reasoning derived from the medical information provided in the question. In addition to common question-answering datasets (Jin et al., 2019; Pal et al., 2022; Jin et al., 2021), we collect real-world clinical questions from The Lancet, the New England Journal of Medicine (NEJM), and Medbullets (Chen et al., 2024) to better assess the clinical utility of LLMs. In the clinical suggestion task, the model is required to provide treatment suggestions (Dubey et al., 2023; Li et al., 2023) or diagnostic decisions (Xie et al., 2022; Fansi Tchango et al., 2022) based on patients' information. In the AI Hospital (Fan et al., 2024) and AgentClinic (Schmidgall et al., 2024) datasets, we task the model with serving as a medical agent. Furthermore, in the MedCalc-Bench (Khandekar et al., 2024) dataset, the model is required to perform mathematical reasoning and calculate answers.

- **Multilinguality** is the ability to complete a task when the languages of input instruction and/or output answers are changed to different languages. For example, XMedBench (Wang et al., 2024) dataset requires LLMs to answer medical questions in six languages, including Chinese, Arabic, Hindi, Spanish, Chinese and English. In AI Hospital dataset (Fan et al., 2024), the model is required to serve as an agent using Chinese.

### 3.3 METRICS

In this section, we elaborate on metrics employed in our evaluation pipeline.

- **Accuracy** is used to directly measure the percentage of models' generated answer which exactly match with the ground-truth. We use accuracy for multi-choice question datasets, MedCalc-Bench (Khandekar et al., 2024) dataset, and portions of clinical suggestion and concept recognition datasets where the ground-truth answer is a single word or phrase.

- **F1-score** (Pedregosa et al., 2011) is the harmonic mean of precision and recall. It is employed in datasets where the model is required to select multiple correct answers.

- **BLEU** (Papineni et al., 2002) and **ROUGE** (Lin & Hovy, 2002) are NLP metrics measuring the similarity between the generated respond and the ground-truth. Specifically, we utilize BLEU-1 and ROUGE-1 for all free-form generation tasks in our evaluation.

- **AlignScore** (Zha et al., 2023) is a metric to measure the factual consistency of generated text. In this paper, we use AlignScore for all free-form generation tasks to evaluate the extent of model's hallucination.

- **Mauve** (Pillutla et al., 2021) is a measure of gap between distribution of generated and human-written text. It is employed for all free-form generation tasks.

All metrics range from 0 to 100, and a higher number indicates better quality output from the model.

## 4 EXPERIMENTS

### 4.1 EXPERIMENT DETAILS

**Prompting strategies.** For most datasets, we employ the same prompting strategy as described in previous literature (Wu et al., 2024b; Nori et al., 2023b;a): For knowledge QA tasks, agent tasks, medical calculation tasks, and multilingual-related tasks, we use the direct prompting evaluation

Table 2: **Accuracy** (Acc.) or **F1** results on 4 tasks across 2 aspects. Model performances with * are taken from Wu et al. (2024b) as the reference. We use the gray background to highlight o1 results. And we present the average score (Average) of each metric in the table

| Aspect | Task | Datasets | Metric | o1 | GPT-4 | GPT-3.5 | MEDITRON* (70B) | Llama3* (8B) |
|---|---|---|---|---|---|---|---|---|
| Understanding | Concept Recognition | PMC-Patient (Zhao et al., 2023) | Acc. | 76.4 | 75.7 | 74.4 | 72.2 | **96.0** |
| | | PICO-Participant (Nye et al., 2018) | Acc. | **75.0** | **75.0** | 52.5 | 72.1 | 58.2 |
| | | PICO-Intervention (Nye et al., 2018) | Acc. | 77.5 | 75.0 | 75.0 | 46.6 | **79.1** |
| | | PICO-Outcome (Nye et al., 2018) | Acc. | **67.5** | 65.0 | 60.0 | 51.2 | 58.2 |
| | | ADE Corpus (Gurulingappa et al., 2012) | Acc. | 78.3 | 78.3 | 71.6 | **95.7** | 69.6 |
| | | Average | Acc. | **74.9** | 73.8 | 66.7 | 67.6 | 72.2 |
| | | BC5-disease (Li et al., 2016) | F1 | **69.5** | 63.0 | 38.9 | 1.4 | 25.3 |
| | | BC5Chem (Li et al., 2016) | F1 | **72.2** | 71.2 | 43.1 | 4.2 | 37.9 |
| | | BC4Chem (Savery et al., 2020) | F1 | **73.4** | 65.1 | 32.7 | 2.0 | 19.5 |
| | | Species800 (Pafilis et al., 2013) | F1 | **71.6** | 66.8 | 55.4 | 0.4 | 11.9 |
| | | HoC (Pafilis et al., 2013) | F1 | **76.3** | 59.0 | 59.8 | 23.7 | 38.3 |
| | | Average | F1 | **72.6** | 65.0 | 46.0 | 6.3 | 26.6 |
| Reasoning | Clinical Decision Support | DDXPlus (Fansi Tchango et al., 2022) | Acc. | **64.0** | 56.0 | 41.0 | 29.6 | 33.8 |
| | | SEER (Dubey et al., 2023) | Acc. | **80.0** | 69.6 | 5.0 | 68.3 | 56.1 |
| | | MIMIC4ED -Hospitalization (Xie et al., 2022) | Acc. | **64.0** | 61.0 | 62.0 | 56.3 | 39.1 |
| | | MIMIC4ED -72h ED Revisit (Xie et al., 2022) | Acc. | **59.7** | 58.0 | 53.6 | 48.5 | 9.3 |
| | | MIMIC4ED -Critical Triage (Xie et al., 2022) | Acc. | 61.7 | **66.7** | 58.7 | 45.7 | 8.8 |
| | | MedNLI-Dis. (Romanov & Shivade, 2018) | Acc. | **88.0** | 84.0 | 57.0 | 60.9 | 63.9 |
| | | PUBHEALTH Ver. (Kotonya & Toni, 2020) | Acc. | **76.4** | 75.7 | 74.4 | 32.7 | 63.9 |
| | | Average | Acc. | **70.5** | 67.3 | 50.2 | 48.9 | 39.3 |
| | Knowledge QA | PubMedQA (Jin et al., 2019) | Acc. | 75.0 | 52.8 | 25.4 | 74.4 | 73.0 |
| | | MedQA (Jin et al., 2021) | Acc. | **75.5** | 69.7 | 53.8 | 47.9 | 60.9 |
| | | MedMCQA (Pal et al., 2022) | Acc. | **95.0** | 79.5 | 58.8 | 59.2 | 50.7 |
| | | Medbullets (Chen et al., 2024) | Acc. | **90.6** | 66.9 | 50.7 | - | - |
| | | LancetQA | Acc. | **81.5** | 76.0 | 61.0 | - | - |
| | | NEJMQA | Acc. | **91.2** | 83.5 | 65.0 | - | - |
| | | Average | Acc. | **84.8** | 71.4 | 52.5 | 60.5 | 61.5 |
| | Medical Calculation | MedCalc-Bench (Khandekar et al., 2024) | Acc. | **34.9** | 25.5 | 10.8 | - | - |

method, which is consistent with the settings of these benchmarks. For other tasks derived from MedS-Bench (Wu et al., 2024b), we follow their benchmark settings, leveraging a few-shot (3-shot) prompt strategy with its template shown in Appendix A.1. As officially suggested by OpenAI, common prompting techniques such as Chain-of-Thought (CoT) (Wei et al., 2022) and in-context examples may not boost o1's performance as it has implicit CoT built in. To further validate this claim, we also investigate the effect of several advanced promptings in our evaluation (*e.g.*, CoT, Self-Consistency (Wang et al., 2022), and Reflex (Shinn et al., 2024)), the detailed input instruction formats are in Appendix A.1

**Models for evaluation.** We choose the following models to evaluate: GPT-3.5 (gpt-3.5-turbo-0125)[3], an advanced language model by OpenAI known for its enhanced contextual understanding; GPT-4 (gpt-4-0125-preview) (Achiam et al., 2023), the successor to GPT-3.5 with significant improvements in reasoning and language comprehension; o1 (o1-preview-2024-09-12) (OpenAI, 2024), the lastest LLM model that is capable of performing highly complex reasoning by employing chain-of-thought reasoning. Apart from these close-source models, we have also incorporated two open-source ones in our experiments: MEDITRON-70B (Chen et al., 2023), an LLM trained with medical-centric data and Llama3-8B (Meta, 2024), the latest and strongest open LLM right now.

## 4.2 MAIN RESULT: *Yes!* WE ARE ONE STEP CLOSER TO AN AI DOCTOR 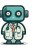

**Enhanced ability of o1 transfers to its clinical understanding.** Given the established results from o1, which underscore its remarkable effectiveness in knowledge and reasoning abilities such as mathematical problem-solving and code generation (OpenAI, 2024), we observe that this superior capability can also be transferred to the specific clinical knowledge understanding. Results presented in Table 2 demonstrate that o1 outperforms other models on the *understanding* aspect in most clinical tasks. We also present these statistics in Figure 1, where we observe that o1 has a larger cover radius

---

[3]https://platform.openai.com/docs/models/gpt-3-5-turbo/

Table 3: **BLEU-1 (B-1)** and **ROUGE-1 (R-1)** results on 3 tasks across 2 aspects. We use the gray background to highlight o1 results. We also present the average score (Average) of each metric

| Aspect | Task | Datasets | o1 | | GPT-4 | | GPT-3.5 | | MEDITRON⋆ (70B) | | Llama3⋆ (8B) | |
|---|---|---|---|---|---|---|---|---|---|---|---|---|
| | | | B-1 ↑ | R-1 ↑ | B-1 ↑ | R-1 ↑ | B-1 ↑ | R-1 ↑ | B-1 ↑ | R-1 ↑ | B-1 ↑ | R-1 ↑ |
| **Understanding** | Text Summary | MIMIC-IV-Ultrasound (Johnson et al., 2023) | **22.2** | **28.8** | 15.9 | 27.0 | 11.0 | 21.1 | 3.8 | 6.1 | 18.1 | 20.0 |
| | | MIMIC-IV-CT (Johnson et al., 2023) | 19.0 | 26.4 | 15.7 | 22.7 | 18.7 | 25.9 | 16.3 | 23.9 | **24.5** | **29.4** |
| | | RCT-Text (Wallace et al., 2021) | 19.5 | 23.4 | 19.5 | 23.4 | **20.6** | **24.2** | 4.0 | 16.4 | 15.4 | 14.6 |
| | | MedQSum (Lee et al., 2021) | **39.2** | **46.8** | 36.3 | 43.0 | 26.5 | 39.6 | 15.6 | 23.1 | 22.5 | 25.1 |
| | | Average | **25.0** | **31.4** | 21.8 | 29.0 | 19.2 | 27.7 | 9.9 | 17.4 | 20.1 | 22.3 |
| | Concept Recognition | HumanDO (Schriml et al., 2019) | **24.9** | **33.1** | 9.7 | 16.2 | 12.2 | 19.4 | 7.7 | 25.4 | 14.9 | 18.8 |
| | | BioLORD (Remy et al., 2024) | **23.0** | **31.8** | 14.7 | 21.8 | 12.8 | 19.1 | 11.8 | 22.7 | 8.9 | 14.6 |
| | | Average | **24.0** | **32.5** | 12.2 | 19.0 | 12.5 | 19.3 | 9.8 | 24.1 | 11.9 | 16.7 |
| **Reasoning** | Clinical Decision Support | EBMS (Mollá & Santiago-Martinez, 2011) | **16.2** | **20.4** | 12.0 | 16.3 | 15.4 | 19.4 | 11.6 | 15.8 | 16.5 | 16.5 |
| | | PUBHEALTH Exp. (Kotonya & Toni, 2020) | 15.8 | **23.6** | 15.1 | 22.0 | 16.6 | 23.6 | 6.1 | 8.7 | **16.8** | 20.3 |
| | | ChatDoctor (Li et al., 2023) | 12.2 | **27.6** | 20.9 | 4.7 | 14.0 | 27.0 | - | - | - | - |
| | | MedNLI-Gen. (Romanov & Shivade, 2018) | **17.0** | **26.0** | 16.9 | 25.8 | 10.0 | 18.3 | 4.4 | 14.1 | 21.3 | 22.8 |
| | | Average | 15.3 | **24.4** | **16.2** | 17.2 | 14.0 | 22.1 | 7.4 | 12.9 | 18.2 | 19.9 |

Table 4: **AlignScore** and **Mauve** results on 3 tasks across 2 aspects

| Aspect | Task | Datasets | AlignScore ↑ | | | Mauve ↑ | | |
|---|---|---|---|---|---|---|---|---|
| | | | o1 | GPT-4 | GPT-3.5 | o1 | GPT-4 | GPT-3.5 |
| **Understanding** | Text Summary | MIMIC-IV-Ultrasound (Johnson et al., 2023) | 27.5 | **30.9** | 23.6 | 6.1 | **7.4** | 7.3 |
| | | MIMIC-IV-CT (Johnson et al., 2023) | **14.4** | 13.3 | 13.8 | 0.4 | **0.5** | **0.5** |
| | | RCT-Text (Wallace et al., 2021) | 4.9 | 4.9 | **5.7** | **3.1** | 2.7 | 11.9 |
| | | MedQSum (Lee et al., 2021) | 34.5 | **37.1** | 13.6 | 42.1 | **52.7** | 0.6 |
| | | Average | 20.3 | **21.6** | 14.2 | 12.9 | **15.8** | 5.1 |
| | Concept Recognition | HumanDO (Schriml et al., 2019) | **17.5** | 5.5 | 5.2 | **8.2** | 0.4 | 0.4 |
| | | BioLORD (Remy et al., 2024) | 13.0 | **19.0** | 17.9 | **51.6** | 4.2 | 1.1 |
| | | Average | **15.3** | 12.3 | 11.6 | **29.9** | 2.3 | 0.8 |
| **Reasoning** | Clinical Decision Support | EBMS (Mollá & Santiago-Martinez, 2011) | **9.0** | 6.6 | 5.7 | **19.5** | 1.9 | 2.3 |
| | | PUBHEALTH Exp. (Kotonya & Toni, 2020) | 14.8 | **19.0** | 17.9 | **2.1** | 0.8 | 1.1 |
| | | ChatDoctor (Li et al., 2023) | **26.5** | 20.4 | 16.6 | **0.7** | 0.5 | 0.6 |
| | | MedNLI-Gen. (Romanov & Shivade, 2018) | 6.8 | **9.7** | 2.5 | **5.3** | 4.5 | 0.9 |
| | | Average | **14.3** | 13.9 | 10.7 | **6.9** | 1.9 | 1.2 |

across various medical datasets. For instance, on 5 concept recognition datasets that use F1 as the metric, o1 outperforms both GPT-4 and GPT-3.5 by an average of 7.6% and 26.6%, respectively (*i.e.*, 72.6% *vs.* 65.0% *vs.* 46.0%), with a notable 24.5% average improvement on the widely used BC4Chem dataset.

Additionally, on the summarization task in Table 3, o1 achieves a 2.4% and 3.7% increase in ROUGE-1 score over GPT-4 and GPT-3.5 (*i.e.*, 31.4% *vs.* 29.0% *vs.* 27.7%), demonstrating its enhanced capacity for real-world clinical understanding. This improved performance confirms that advancements in general NLP capabilities for LLMs can effectively translate to enhanced model understanding in the medical domain.

**The o1 model demonstrates strong reasoning in clinical diagnosis scenarios.** On the reasoning aspect, o1 takes a significant step forward in demonstrating its advantages in real-world diagnostic situations. In our newly constructed challenging QA tasks, NEJMQA and LacentQA, o1 showcases an average accuracy improvement of 8.9% and 27.1% over the performance of GPT-4 (79.6%) and GPT-3.5 (61.5%) on the respective datasets (Table 2). Another noteworthy improvement in o1 is its capacity for mathematical reasoning, elevating the baseline of MedCalc-Bench to 34.9%, which surpasses GPT-4 by a significant 9.4%. In more complex reasoning scenarios that involve multi-turn conversations and environmental simulations, o1 outperforms both GPT-4 and GPT-3.5 on the AgentClinic benchmark, achieving accuracy gains of at least 15.5% and 10% with scores of 45.5% and 20.0% on its MedQA and NEJM subsets, respectively. These observations serve as compelling evidence of o1's competence in complex real-world diagnosis and clinical utility scenarios.

In addition to delivering higher accuracy, o1 provides more concise and straightforward answers. In the example illustrated in Figure 4, o1 generates shorter interpretations while offering the correct answer. In contrast, GPT-4 tends to generate hallucinated explanations alongside incorrect answers. We believe o1's improvement in both knowledge and reasoning is primarily attributed to the enhanced

**Question**: A 2-month-old infant with a 1-month history of a generalised persistent blistering rash (figure) and a 4-day history of vomiting and poor oral intake attended our hospital with her mother. What is the most likely cause of the rash that appears at 1 month of age and resolves naturally by 2 months?
**Options**:
A) Congenital candidiasis  B) Congenital CMV infection  **C) Congenital syphilis**  D) Herpes simplex infection

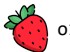 o1

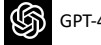 GPT-4

**Answer:** C) Congenital syphilis ✔
**Reasoning:**
The infant presented with a generalized persistent blistering rash that began at 1 month of age and resolved by 2 months. Among the options, congenital syphilis is known to present with bullous skin lesions (pemphigus syphiliticus) in infants, typically appearing around this age. These lesions can be generalized and involve the palms and soles. **The vomiting and poor oral intake could be associated systemic symptoms of congenital syphilis.** The other options are less likely: **congenital candidiasis usually presents earlier and isn't characterized by a generalized blistering rash; congenital CMV typically presents at birth with petechial rashes; neonatal herpes simplex infection usually presents within the first few weeks of life.**

**Answer:** A) Congenital candidiasis ✘
**Reasoning:**
To determine the most likely cause of the rash that **appears at 1 month of age and resolves naturally by 2 months in a 2-month-old infant**, we need to consider the clinical presentation and natural course of the diseases listed in the options.
**Option A: Congenital candidiasis typically presents shortly after birth and is not known to spontaneously resolve within a month.** It usually requires antifungal treatment.
**Option B:** Congenital CMV infection can present with a rash, **but it is unlikely to resolve naturally by 2 months. It often comes with other systemic symptoms** like hepatosplenomegaly, petechiae, and neurological findings.
**Option C:** Congenital syphilis can cause a rash, but it often persists and can be accompanied by other systemic signs such as rhinitis, hepatosplenomegaly, and bone abnormalities. **The rash does not typically resolve on its own without treatment.**
**Option D:** Herpes simplex infection may present with a generalized blistering rash in neonates, and some cases can resolve on their own. **However, herpes simplex infection can be severe and often requires antiviral treatment,** especially with systemic or CNS involvement, which can lead to vomiting and poor oral intake.
Given the data, **the best fit for a self-resolving rash by 2 months of age among these options is congenital candidiasis**, as it may manifest early and is less likely to cause severe systemic illness leading to hospitalization mainly for rash concerns. The other conditions generally do not have a natural history of resolution without medical intervention.

Figure 4: Answers from o1 and GPT-4 on a question from LancetQA. o1 provides a more concise and accurate reasoning process compared to GPT-4.

Table 5: **Accuracy** of LLMs on two agentic benchmarks

| Tasks | AI Hospital (Fan et al., 2024) | | | | | AgentClinic (Schmidgall et al., 2024) | |
|---|---|---|---|---|---|---|---|
| | Symp. | Medical Exam. | Diagnostic Results | Diagnostic Rationales | Treatment Plan | MedQA | NEJM |
| o1 | **67.0** | 43.4 | **45.1** | 45.1 | **39.9** | **45.5** | **20.0** |
| GPT-4 | 66.7 | **45.0** | 44.2 | **45.8** | 38.2 | 30.4 | 10.0 |
| GPT-3.5 | 62.0 | 40.7 | 35.8 | 36.3 | 24.7 | 25.2 | 7.5 |

data and infrastructure employed during the training process (*e.g.*, CoT data and the reinforcement learning technique).

These results together provide a positive answer to the question we raised in this paper: *Yes!* We are getting closer to an automatic AI doctor with the latest o1 model.

### 4.3 FURTHER ANALYSIS

**No model excels across all tasks in the medical domain.** Table 2 and Table 3 indicate that, for now, there are always trade-offs (even under the same metric) to be made when selecting a model to use in the medical domain. One example is the clinical decision support task in Table 2, o1 outperforms both GPT-4 and GPT-3.5 on most datasets, but lags far behind GPT-4 on the MIMIC4ED-Critical Triage dataset by 5% in accuracy. Interestingly, we also found the recent released open LLM—Llama3 takes a lead in PMC-Patient and PICO-Intervention datasets with an unexpected 19.6% accuracy gap between o1 and Llama3 on PMC-Patient (76.4% *vs.* 96.0%). Nevertheless, o1 comes close to being the best in most situations, it boasts a leading position across datasets in clinical decision support, knowledge QA, and medical calculation. This claim is supported by the average result over 19 dataset accuracy in Table 2 and Figure 2: o1 (74.3%) > GPT-4 (68.1%) > GPT-3.5 (53.2%)

**Advanced prompting can partially help models trained with CoT data.** o1 was released using chain-of-thought (CoT) data embedding in the training process; however, we found that applying the CoT prompting still enhances o1's performance on knowledge QA tasks in medicine, as shown in Table 6. The table reveals an average boost of 3.18% over the original 83.6% accuracy of o1. While this improvement is not as significant as with GPT-4, CoT proves to be a promising way for guiding o1 in medical tasks. However, when it comes to other fancy promptings, such as self-consistency (SC) (Wang et al., 2022) and reflex (Shinn et al., 2024), this conclusion may not stand still. We

Table 6: **Accuracy** results of model results with/without CoT prompting on 5 knowledge QA datasets

| Datasets | `o1` | `o1` (CoT) | `GPT-4` | `GPT-4` (CoT) |
|---|---|---|---|---|
| PubMedQA (Jin et al., 2019) | 75.0 | **75.2** | 52.8 | **62.2** |
| MedQA (Jin et al., 2021) | 95.0 | **95.2** | 79.5 | **86.1** |
| MedMCQA (Pal et al., 2022) | 75.5 | **81.9** | 69.7 | **72.6** |
| LancetQA | 81.5 | **85.5** | 76.0 | **81.5** |
| NEJMQA | 91.2 | **96.3** | 83.5 | **86.4** |

Table 7: **Accuracy** ablation results of using different promptings using `o1` on our LancetQA

| CoT | SC | Reflex | Accuracy |
|---|---|---|---|
|  |  |  | 81.5 |
| ✓ |  |  | 85.5 |
| ✓ | ✓ |  | 84.5 |
| ✓ |  | ✓ | 61.0 |

Table 8: **Accuracy** of models on the multilingual task, XmedBench (Wang et al., 2024)

| Models | English | Chinese | French | Spanish | Arabic | Hindi | Average |
|---|---|---|---|---|---|---|---|
| `o1` | **76.4** | **80.2** | **95.4** | **95.0** | **74.9** | **89.3** | **85.2** |
| `GPT-4` | 75.7 | 61.0 | 89.4 | 91.2 | 60.8 | 76.3 | 75.7 |
| `GPT-3.5` | 72.0 | 47.4 | 58.9 | 74.2 | 39.7 | 32.5 | 54.1 |
| `Meditron-70B*` | 58.7 | 44.3 | 53.3 | 59.7 | 19.3 | 31.3 | 44.4 |

witness an average performance decline of 12.8% using these two strategies compared to only CoT on LancetQA (Table 7).

**Hallucination remains a significant challenge.** We use AlignScore (Zha et al., 2023) to evaluate hallucination in LLMs. In Table 4, the `o1` model demonstrates a 1.3% decrease in AlignScore compared to `GPT-4` across five text summarization datasets. Moreover, the overall improvements of `o1` across three tasks (Table 4) in AlignScore significantly lag behind those of other evaluation metrics—averaging 0.7 in AlignScore compared to 9.9 in Mauve relative to `GPT-4`. This indicates that `o1` is still susceptible to language hallucination, highlighting that such problem remains a persistent challenge in LLMs.

**`o1` struggles in reasoning over complex multilingual tasks.** Advanced LLMs are expected to demonstrate equivalent reasoning abilities to languages other than English. However, as `o1` consistently outperforms other models in multilingual QA tasks: `o1` (85.2%) > `GPT-4` (75.7%) > `GPT-3.5` (54.1%) on average (Table 8), it falls short in a much more complex Chinese agent benchmark in Table 5—showing a 1.6% accuracy drop in the medical examinations scenario over `GPT-4` (43.4% *vs.* 45.0%), leaving its multilingual reasoning in complex situations to be desired. This interesting outcome might be attributed to the lack of multilingual CoT data during `o1`'s training, as learning complex reasoning routes generally requires more efforts than plain instructions in the few-shot paradigm (Kim et al., 2023; Singh et al., 2024). We present a failure example of `o1` on AI Hospital in Figure 5. We identified instances of mixed language output in the generation from the doctor, which contribute to the suboptimal performance of `o1` in this context.

**LLMs are facing biased judgement using different metrics.** Choosing different metrics can lead to varied results of LLM evaluation (Liang et al., 2022), in our experiments, we observe a similar unaligned trend even leveraging traditional NLP metrics such as BLEU-1, ROUGE-1, and Mauve. In most cases from Table 3, `o1` surpasses `GPT-4` in both two traditional reference-based measurements (*i.e.*, BLEU-1, ROUGE-1) on average. One exception arises in the BLEU-1 comparison for clinical suggestion tasks. While `o1` significantly triumph over `GPT-4` in ROUGE-L (24.4% *vs.* 17.2%), it surprisingly underperforms in BLEU-1: `o1` (15.3) < `GPT-4` (16.2). When considering Mauve scores, although `o1` consistently surpasses `GPT-4` in both averaged BLEU-1 and ROUGE-1 for text summarization tasks, it still falls short by 2.9 points in Mauve, even when evaluated on the same

output texts. A similar anomaly can also be observed in the comparison between accuracy and F1 score. While `Llama3` significantly outperforms `o1` in accuracy on two concept recognition datasets, it consistently falls behind `o1` in F1 on the same cases. These findings underscore the urgent need to identify or devise more reliable metrics for modern LLMs.

## 5 DISCUSSION

**What adverse impacts does `o1` bring?** The model `o1` has made significant strides in both general NLP and the medical domain—as demonstrated in this paper. But what adverse impacts does `o1` have on users compared to the previous generations of LLMs? While embedding the Chain of Thought (CoT) process during generation by default requires more time (OpenAI, 2024), what exactly distinguishes `o1` from other OpenAI models? In Table 10, we see that `o1` has more than $2\times$ and $9\times$ longer decoding time cost on four medical tasks compared to `GPT-4` and `GPT-3.5`, respectively (13.18s *vs*. 6.89s *vs*. 1.41s). This increased decoding time can lead to significant waiting periods when handling complex tasks.

Additionally, `o1` does not always outperform other models, with inconsistent performance across different tasks. For instance, in the concept recognition task detailed in Table 2, `o1` underperforms compared to other LLMs on half of the datasets. This discrepancy may relate to recent findings suggesting that CoT data is most advantageous in more complex reasoning tasks (Sprague et al., 2024). However, in tasks that do not require complex reasoning, such as concept recognition, `o1` does not have significant advantages over them.

**Rethinking evaluation metrics for stronger LLMs.** Traditional evaluation metrics like BLEU and ROUGE, which rely on n-gram overlap, have long been criticized for their limitations in capturing the quality of generated text, particularly for LLMs. As a result, using models like GPT-4 as evaluators, *i.e.*, "LLM-as-a-judge", has gained popularity for assessing the outputs of other models. However, this approach may not be valid when applied to the most advanced models such as `o1`, as GPT-4 is even less capable and thus may produce less reliable evaluation. This is especially true for specialized domain like medicine. Therefore, there is a growing need to develop more robust and nuanced evaluation metrics that can better assess the performance of state-of-the-art LLMs in complex scenarios.

**Call for reliable prompting techniques for future LLMs.** As noted in Section 4.3, not all advanced prompting techniques positively impact `o1`'s performance. As future LLMs like `o1` may continue to evolve with internal prompts for efficient user instruction, new prompting methods should consider their adaptability to existing strategies. One potential exploration could be the integration of two prompting strategies (Wang et al., 2022; Zheng et al., 2024).

**Limitations.** While we conduct comprehensive evaluations in the medical domain on understanding, reasoning, and multilingual capabilities, there are many other dimensions to consider such as safety (Han et al., 2024) and we leave them for future work. Additionally, we leave more advanced prompting techniques such as retrieval augmented generation (RAG) (Lewis et al., 2020) for future work, which may enhance the factuality and mitigate hallucination. It is worth noting that current GPT-like models may still underperform BERT-based specialists in classification tasks (Nori et al., 2023b). However, we focus on GPT-like generalists in this paper due to their greater flexibility as zero-shot learners.

## 6 CONCLUSION

This preliminary study assesses 3 important aspects across 35 existing and 2 novel medical datasets using the latest `o1` model. It marks the first step towards a holistic evaluation of `o1` in medicine, and we present our initial results, analysis, and discussion over the benchmark. The findings provide convincing evidence that `o1` is narrowing the gap between AI and human doctors, shaping the vision of an ideal AI doctor closer to reality.

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

# A  APPENDIX

## A.1  PROMPTING STRATEGIES

---

**Base Prompt for MCQ.**

Question:
{question}

Options:
A) ......
B) ......
.......

{Format Constraint}

---

**Format Constraint Examples for MCQ.**

**Default:**
Answer only with the option index such as A/B/C/D in plain text.
**True/False Statement Questions:**
Answer only with Yes/No in plain text.

---

**Few-Shot Prompt.**

Case1: ...
Case2: ...
Case3: ...
...
{Manually Written Definitions}
Please learn from the few-shot cases to see what content you have to output.
{Input Case}

---

**CoT Format Constraint.**

Reason step-by-step before answering. {Base Format Instruction}. Your final output should strictly follow this format:
⟨Reason⟩{your step-by-step reasoning}⟨/Reason ⟩⟨Answer⟩{your answer}⟨/Answer⟩

---

**Self Consistency.**

Given the following question and the {n_sample} answers, please select the most consistent response with other answers and the question. {Base Format Constraint} in strictly this format: ⟨Answer⟩{your final answer}⟨/Answer⟩.

# Question: {Base Prompt with CoT}

# Answer 1:
{Model Answer 1}

# Answer 2:
{Model Answer 2}

# Answer 3:
{Model Answer 3}

---

# Response:
{Model Response}

Please review the answer above and criticize on where might be wrong. If you are absolutely sure it is correct, output 'True'.

**Prompt for Reflected Answer Generation for Reflex.**

{Base Prompt with CoT Format Constraint}

# Original Answer:
{Model Answer}

# Critic:
{Model Critic}

Given previous attempts and feedback, carefully consider where you could go wrong in your latest attempt. Using insights from previous attempts, try to solve the task better.

**Prompt for Final Answer Generation for Reflex.**

{Base Prompt with CoT Format Constraint}

# Answer 1:
{Reflected Answer 1}

# Answer 2:
{Reflected Answer 2}

# Answer 3:
{Reflected Answer 3}

Please summarize the previous attempts and feedback and provide a final answer. {Base Format Constraint} in strictly this format: ⟨Answer⟩{your final answer}⟨/Answer⟩.

## A.2    DETAILS ABOUT DATASETS

In this paper, we present a summary of 36 medical-related datasets spanning 6 distinct tasks, as outlined in Table 1. Notably, the inclusion of commercial models, particularly `o1`, leads to significant costs and response latency. To address this, for some tasks we randomly sampled a subset of test cases, which are detailed below.

**Concept Recognition**

- **BC4Chem** (Savery et al., 2020) is a dataset comprising 10,000 PubMed abstracts with 84,355 chemical entity mentions, manually annotated by expert chemistry literature curators. The task is to extract chemical names from the given abstracts. For evaluation, we randomly sample 300 instances from the test set.

- **BC5Chem** and **BC5Disease** are from BC5CDR (Li et al., 2016), a widely-used resource in biomedical natural language processing, annotated for chemical and disease entities and their relationships. Following MedS-Bench (Wu et al., 2024b), BC5CDR is split into 2 datasets: chemical name extraction and disease name extraction. For evaluation, we randomly sample 300 instances from each task's test set.

- **Species800** (Pafilis et al., 2013) comprises 800 PubMed abstracts with annotated organism mentions. The task is to extract organism names from the given abstracts. For evaluation, we randomly sample 300 instances from the test set.

- **HoC** (Baker et al., 2016) is a specialized dataset containing 1,852 PubMed publication abstracts, expertly annotated according to a taxonomy of cancer hallmarks. The task is to classify the hallmarks of cancer based on the given biomedical publication abstracts. For evaluation, we use the entire test set consisting of 158 instances.

- **HumanDiseaseOntology** (Schriml et al., 2019) is a database providing consistent, reusable, and sustainable descriptions of human disease terms, phenotype characteristics, and related medical vocabularies. The task is to explain specified medical professional entities, with the database descriptions serving as ground truth. For evaluation, we randomly sample 300 instances.

- **BioLORD** (Remy et al., 2024) comprises pairs of biomedical concept names and descriptions. The task is to elaborate on concise concepts by generating long, detailed definitions. For evaluation, we randomly sample 300 instances.

- **PMC-Patient** (Zhao et al., 2023) is a collection of 167,000 patient summaries extracted from case reports in PubMed Central (PMC), annotated with basic patient information. The task is to extract patient gender and age information from given clinical texts. For evaluation, we randomly sample 300 instances.

- **PICO-Participant**, **PICO-Intervention** and **PICO-Outcome** are three datasets derived from PICO (Nye et al., 2018), consisting of 5,000 abstracts from medical articles on randomized controlled clinical trials. The tasks involve extracting information about study participants, interventions, and outcomes from given sentences. For evaluation, we use the entire test set of 43 instances for each task.

- **ADE Corpus** (Gurulingappa et al., 2012) provides information on drugs and their corresponding adequate doses within sentences. The task is to extract the dosage levels of specified drugs from given sentences and drug names. We use the dataset prompted by Super-Instruction with a 9:1 ratio for instruction tuning and evaluation. The test set consists of 23 instances.

**Text Summary**

- **MIMIC-IV-CT** and **MIMIC-IV-Ultrasound** (Johnson et al., 2023; Wallace et al., 2021) are subsets of MIMIC-IV Report, a large deidentified medical dataset of patients admitted to the Beth Israel Deaconess Medical Center. The task is to summarize radiology reports, treating the impression part as a general summary of the findings. Following (Wu et al., 2024b), we randomly sampled 500 cases from body region part of Chest CT and 100 cases from ultrasound modality for evaluation.

- **RCT-Text** (Wallace et al., 2021) is a dataset for summarizing medical evidence from clinical studies in literature reviews. The task is to output the primary conclusions of each study given the titles and abstracts. For evaluation, we randomly sample 100 instances.

- **MedQSum** (Lee et al., 2021) is derived from a large database of de-identified health-related data. The task is to generate a summary of detailed findings from imaging diagnostic reports, with the conclusion of the note serving as ground truth. For evaluation, we randomly sample 100 instances.

**Knowledge QA**

- **MedQA** (Jin et al., 2021) is a collection of medical multiple-choice questions in English. We use the 4-option English version with the official split. The test set contains 1273 samples.

- **PubMedQA** (Jin et al., 2019) is an English question-answering dataset based on PubMed abstracts. The task is to answer research questions with yes/no/maybe. We use the PQA-L subset as the test set, containing 1000 samples.

- **MedMCQA** (Pal et al., 2022) is a large-scale English multiple-choice question-answering dataset from AIIMS & NEET PG entrance exams. We use the official test split containing 4183 questions, each with 4 choices.

- **LancetQA** and **NEJMQA** are datasets curated from The Lancet and the New England Journal of Medicine case challenges, focusing on patient diagnosis based on symptoms. We use 200 samples for LancetQA and 100 samples for NEJMQA.

- **Medbullets** (Chen et al., 2024) is a dataset curated from the Medbullets online platform, comprising 308 USMLE Step 2&3 style questions. Each question includes a case description, four answer choices, and an explanation.

**Clinical Decision Support**

- **DDXPlus** (Fansi Tchango et al., 2022) is a dataset for Automatic Symptom Detection and Automatic Diagnosis systems, featuring synthesized patient data. The task is to make diagnostic decisions based on dialogues. For evaluation, we randomly sample 300 instances.

- **SEER** (Dubey et al., 2023) is a treatment planning dataset based on the Surveillance, Epidemiology, and End Results breast cancer databases. The task is to recommend treatment plans from five types. For evaluation, we randomly sample 300 instances.

- **MIMIC4ED-Hospitalization**, **MIMIC4ED-72h ED Revisit**, and **MIMIC4ED-Critical Triage** are datasets from the MIMIC4ED Benchmark (Xie et al., 2022) for predicting clinical outcomes in emergency medicine. For each dataset, we randomly sample 300 instances for evaluation.

- **MedNLI-Dis.** (Discriminative) and **MedNLI-Gen.** (Generative) are derived from MedNLI (Romanov & Shivade, 2018), a natural language inference dataset for the clinical domain. The dataset involve discriminative and generative entailment based on clinical premises. For each task, we randomly sample 300 instances for evaluation.

- **EBMS** (Mollá & Santiago-Martinez, 2011) is a justification verification dataset. We use the entire test set of 304 instances for evaluation.

- **PUBHEALTH Exp.** (Explanation) (Kotonya & Toni, 2020) requires models to provide explanations for specified claims using supporting material from given paragraphs. For evaluation, we randomly sample 300 instances.

- **PUBHEALTH Ver.** (Verification) (Kotonya & Toni, 2020) is a fact verification task where models determine if a claim contradicts evidence in a given paragraph. For evaluation, we randomly sample 300 instances.

- **Chatdoctor** (Li et al., 2023) is based on 100K patient-physician conversations from an online medical consultation website[4]. The task involves engaging in medical consultations based on this data.

**Agent**

- **AI Hospital** (Fan et al., 2024) is a multi-agent framework simulating medical interactions in Chinese. It includes Patient, Examiner, Chief Physician, and Doctor agents, with 506 cases from diverse departments. The task involves simulating clinical scenarios through dialogue. Evaluation uses Chief Physician's 1-4 scale scoring across five dimensions: symptoms, examinations, diagnostic results, rationales, and treatment plan. 200 cases are sampled for evaluation.

- **AgentClinic** (Schmidgall et al., 2024) is a clinical environment benchmark with 107 patient agents from MedQA and 15 multimodal agents from NEJM challenges. The task is patient diagnosis through dialogue and data collection. Evaluation considers diagnostic accuracy and patient perception metrics in biased scenarios.

**Medical Calculation**

- **MedCalc-Bench** (Khandekar et al., 2024) evaluates LLMs' medical calculation abilities using 1,047 instances across 55 tasks. It requires computing medical values from patient notes and questions. Evaluation compares LLM outputs to ground truth, with exact matches for rule-based and 5% tolerance for equation-based calculators.

**Multilinguality**

- **XMedBench** (Wang et al., 2024) is a multilingual medical benchmark in six languages: English, Chinese, Hindi, Spanish, French, and Arabic. It uses multiple-choice questions from various sources, including translated versions for Arabic and Hindi. The task evaluates LLMs' medical knowledge across languages, using accuracy as the primary metric.

- **AI Hospital** (Fan et al., 2024) is a multi-agent framework simulating medical interactions in Chinese. We also include this dataset into the multilinguality aspect because it is in Chinese.

---

[4]www.healthcaremagic.com

## A.3 Model-based Evaluation

As discussed in Section 5, Rethinking evaluation metrics for stronger LLMs, we also explore using techniques such as "LLM-as-a-judge" to assess the quality of generated outputs. Table 9 shows that `o1` achieves nearly the same score as `GPT-4` and outperforms `GPT-3.5` (i.e., 3.3% vs. 3.3% vs. 3.0%), which contrasts with the traditional evaluation metrics in Table 3. This indicates that the "LLM-as-a-judge" method may be unreliable when applied to advanced models like `o1`, as GPT-4, being less capable, may provide less accurate evaluations. This limitation is particularly evident in specialized domains such as medicine. The prompt used for "LLM-as-a-judge" is shown in Appendix A.3.

Table 9: GPT Evaluation Score Comparison

| Task | Datasets | GPT Score ↑ | | |
|------|----------|------|-------|--------|
| | | `o1` | `GPT-4` | `GPT-3.5` |
| **Text Summarization** | medqsum | 4.1 | 3.8 | 4.1 |
| | RCT-Text | 3.2 | 3.2 | 3.1 |
| | MIMIC-IV-Ultrasound | 3.8 | 3.8 | 3.4 |
| | MIMIC-IV-CT | 3.8 | 3.8 | 3.7 |
| **Clinical Suggestion** | MedNLI-Generative | 2.3 | 2.4 | 2.5 |
| | EMBS Justification Ver. | 3.1 | 3.0 | 3.0 |
| | PUBHEALTH Exp. | 3.0 | 3.3 | 3.2 |
| | Do Entity Exp. | 3.7 | 3.6 | 3.3 |
| | BioLORD Concept Exp. | 3.3 | 3.3 | 3.0 |
| | ChatDoctor | 2.5 | 2.6 | – |
| **Average** | | 3.3 | 3.3 | 3.3 |

---

**Prompt for LLM-as-a-judge.**

You are a senior medical expert. Please evaluate the quality of the medical text material provided by medical interns based on the expert medical text material as a reference answer. The quality is divided into five levels:

5. The assistant result completely matches the reference.
4. The assistant result is generally consistent with the reference, with only a small part of omissions or errors.
3. The assistant result partially matches the reference, but there are some omissions and errors.
2. The assistant result is mostly inconsistent with the reference, with many omissions and errors.
1. The assistant result is completely inconsistent with the reference.

{Input Medical Questions}
Assistant Result: {Result}
Reference Answer: {Reference}

Please note:
(1) Focus on the factual content of the medical answers, without concern for style, grammar, punctuation, and non-medical content. (2) Your response should be in the format. Rating: (int)

---

## A.4 Decoding Time

We evaluated the model's time cost and the average number of decoding tokens across various tasks, including Knowledge QA, Clinical Decision Support, Text Summary, and Concept Recognition. For each task, we select a representative dataset and perform inference on 50 samples. The time and decoded tokens are then averaged to obtain the results for each response, as illustrated in Table 10. The decoding time for `o1` is significantly higher than both `GPT-4` and `GPT-3.5`, taking more than double the time of `GPT-4` and over nine times that of `GPT-3.5` across four medical tasks (13.18s compared to 6.89s and 1.41s, respectively).

Table 10: Model **time** cost and averaged number of decoding tokens for 4 datasets across 4 tasks

| Task | Dataset | Model | Time (s) | Prompt Tokens | Completion Tokens | Reasoning Tokens | Total Tokens |
|---|---|---|---|---|---|---|---|
| Knowledge QA | MedQA | o1 | 11.13 | 247.78 | 953.42 | 924.16 | 1201.20 |
| | | GPT-4 | 0.83 | 236.20 | 9.26 | 0 | 245.46 |
| | | GPT-3.5 | 0.52 | 236.20 | 10.02 | 0 | 246.22 |
| Clinical Decision Support | ChatDoctor | o1 | 11.40 | 122.64 | 1127.44 | 83.84 | 1250.08 |
| | | GPT-4 | 18.88 | 124.24 | 509.28 | 0 | 633.52 |
| | | GPT-3.5 | 2.40 | 124.24 | 150.10 | 0 | 274.34 |
| Text Summary | MIMIC-IV | o1 | 20.56 | 1305.54 | 1080.54 | 1057.28 | 1373.32 |
| | | GPT-4 | 6.26 | 1254.84 | 162.68 | 0 | 1417.52 |
| | | GPT-3.5 | 2.02 | 1254.84 | 159.94 | 0 | 1414.78 |
| Concept Recognition | BC5Chem | o1 | 9.62 | 292.78 | 1080.54 | 1057.28 | 1373.32 |
| | | GPT-4 | 1.60 | 297.24 | 19.64 | 0 | 316.88 |
| | | GPT-3.5 | 0.68 | 297.24 | 12.80 | 0 | 310.04 |

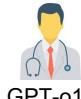

GPT-o1

您好，有哪里不舒服？

我并没有什么自觉症状，只是体检的时候发现了胆囊里有个息肉，并且这一年里它慢慢长大了。

patient

请问您最近的检查显示，胆囊息肉的大小是多少？

最近的检查显示，胆囊息肉较前增大。

请问目前胆囊息肉的具体大小是多少？

抱歉，我不知道具体的大小是多少。检查员能告诉我胆囊息肉的具体大小吗？

了解，胆囊息肉的大小对于评估风险和制定治疗方案非常重要。如果您无法获取具体的大小信息，建议您联系进行检查的医院或医生，获取详细的检查报告。这样我们才能更准确地评估您的情况。一般而言，**胆囊息肉小于1厘米且无症状**，可以选择**定期观察，每6-12个月复查一次**；如**果息肉大于1厘米**，或者**在短时间内迅速增大**，则需要考虑**手术切除**，以防止潜在的恶变风险。在您获取到具体的检查结果后，我们可以进一步讨论最适合您的治疗方案。

**症状**：(1) 无自觉症状。**辅助检查**：(1) 体检时发现胆囊内有息肉。(2) 最近的检查显示胆囊息肉较前增大。**诊断结果**：胆囊息肉。**诊断依据**：(1) 影像学检查发现胆囊内存在息肉样病变。(2) 息肉在一年内出现增大趋势。**治疗方案**：(1) 建议尽快获取胆囊息肉的具体大小和影像学特征，以评估恶变风险。(2) 根据息肉的大小和生长速度，考虑定期随访观察或进行手术切除。一般来说，息肉 larger than 1 厘米或迅速增大者，建议手术治疗。

Figure 5: **Failure case** of o1 on AI Hospital. The model struggles with generating the right diagnosis and outputs mixed-language, resulting to its suboptimal performance in this context.

## A.5 CASE STUDY

To demonstrate how the advanced o1 model outperforms previous methods in potential clinical applications, we present comparative case studies. As shown in Figure 6 and Figure 4, when responding to questions from NEJMQA and LancetQA, o1 provides the correct answer with a more concise and precise reasoning process. In contrast, GPT-4 generates a longer reasoning path while leading to an incorrect answer. in Figure 7, when offering diagnosis and treatment based on patient records, o1 delivers a more accurate diagnosis and practical treatment plan, closely aligning with human experts and outperforming GPT-4. These cases demonstrate that o1 more closely resembles an AI Doctor.

**Question**: A 60-year-old left-handed woman presented to the emergency department with pain in her left forearm. The arm was swollen and tender, especially with passive pronation and supination. The overlying skin was intact, and the results of neurovascular examination were normal. Radiographs of the left forearm were performed (upper image, anteroposterior view; lower image, lateral view). The findings should raise concern for which contributory factor?

**Options**:
A) Assault  B) Fall  C) Malignancy  D) Osteoporosis  E) Repetitive strain

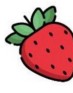

o1

**Answer:** A) Assault ✅
**Reasoning:**
The patient is a 60-year-old woman presenting with swelling and tenderness in the left forearm, which is exacerbated by passive pronation and supination. The overlying skin is intact, and neurovascular examination is normal, suggesting a closed fracture without neurovascular compromise. Radiographs were performed to assess for possible fractures.

In this age group, osteoporosis is common and can lead to fractures from minimal trauma. **However, osteoporotic fractures typically occur in the distal radius (Colles' fracture) after a fall on an outstretched hand. The pain with pronation and supination indicates involvement of the forearm bones rather than the wrist.**

An isolated fracture of the ulnar shaft, known as a nightstick fracture, results from a direct blow to the ulna. **This type of injury is classically associated with a defensive action when a person raises their forearm to protect themselves from an overhead strike, often in the context of an assault. The intact skin suggests there was no penetrating injury, aligning with a blunt force trauma.**

Therefore, **the radiographic findings of an isolated ulnar fracture in this clinical context should raise concern for assault as the contributory factor.**

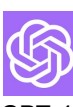

GPT-4

**Answer:** D) Osteoporosis ❌
**Reasoning:**
The patient presents with pain, swelling, and tenderness in her left forearm following some unwitnessed event, with an emphasis on particular pain during passive movements (pronation and supination), but with intact skin and normal neurovascular findings. The key elements to analyze include the patient's age, the nature of the injury, and what can be deduced from the symptoms and radiographic findings without seeing the images.
1. **Assault:** While an assault can lead to trauma presenting with pain and swelling, **there is no specific evidence or context provided that indicates an assault occurred.** This option might be considered if there were additional signs of defensive wounds or if the history provided such information.
2. **Fall:** A fall, especially in a 60-year-old, can easily result in fractures due to decreased bone density that comes with age. This option seems plausible since falls can cause direct trauma leading to the symptoms described. **However, without explicit information about a fall or the mechanism of injury, it cannot be conclusively selected based solely on the symptoms.**
3. **Malignancy:** While malignancy could theoretically present with bone pain or pathological fractures, the acute presentation in the emergency department more closely matches trauma or injury. **A malignancy would likely have a more insidious onset and possibly other systemic symptoms.**
4. **Osteoporosis:** In a 60-year-old woman, osteoporosis is a significant risk factor for fractures, especially with minimal or no trauma. Osteoporosis could make bones more susceptible to fracture from otherwise non-severe impacts. Given the age and gender of the patient, **this is a factor that could contribute to the ease of which a fracture or injury could occur, especially under stress or minor trauma conditions that might not affect a younger individual or someone without osteoporosis.**
5. **Repetitive strain:** This typically presents with soft tissue injuries rather than fractures and is more chronic in nature. **The described** acute presentation and radiographs (implied to show a fracture or some abnormal finding) **do not fit the typical pattern of a repetitive strain injury, which more commonly affects tendons and muscles rather than resulting in acute bony injury.**

Considering these options and reasoning, **Osteoporosis seems the most plausible contributory factor to the concern raised by the radiographs.** It is a common condition affecting older adults, especially women, where decreased bone density increases the risk of fractures from minimal trauma that would not typically cause injury in individuals with healthy bone density.

Figure 6: Comparison of the answers from GPT-o1 and GPT-4 for a question from NEJMQA. o1 provides a more concise and accurate reasoning process compared to GPT-4.

**Patient Record**

**症状：** 患者因胃溃疡复发接受治疗，胃镜检查后病理诊断为胃窦印戒细胞癌。患者有反酸、嗳气，无胃部胀痛、发热寒颤、意识障碍、恶心呕吐、咳嗽咳痰、咯血、心悸胸闷，饮食及二便正常。以"胃窦恶性肿瘤"为诊断收住院。
- 血常规：- 白细胞：3.20*10^9/L - 红细胞：3.68*10^12/L - 血红蛋白：132g/L - 红细胞压积：39.20% - 平均血红蛋白浓度：337g/L
- 影像学检查：- CT：胃壁局部增厚，厚约1.7cm
- 病理检查：- 胃窦印戒细胞癌

**Symptoms:** The patient was being treated for a recurrent gastric ulcer. After gastroscopy, the pathological diagnosis was gastric antrum signet ring cell carcinoma. The patient had acid reflux and belching, but no stomach distension, fever and chills, consciousness disorders, nausea and vomiting, cough and sputum, hemoptysis, palpitations and chest tightness. Diet and bowel movements were normal. He was admitted to the hospital with the diagnosis of "malignant tumor of the gastric antrum".
- Blood routine: - White blood cells: 3.20*10^9/L - Red blood cells: 3.68*10^12/L - Hemoglobin: 132g/L - Hematocrit: 39.20% - Mean hemoglobin concentration: 337g/L
- Imaging tests: - CT: local thickening of the gastric wall, about 1.7cm thick
- Pathology: - Signet ring cell carcinoma of the antrum of the stomach

**Human Expert**

**诊断：** 1. 胃窦印戒细胞癌 2. 胃壁肿物
**治疗：** 患者入院后进行术前准备，无手术禁忌症，行胃癌根治术。术中未发现肝脏、盆腔转移，无腹水，胃窦部肿物直径约2cm，胃体部浆膜下肿物0.5cm*1.3cm，清除相关淋巴结。术后给予抗感染、抑制胃酸分泌、补液等治疗

**Diagnosis:** 1. gastric antral signet ring cell carcinoma 2. gastric wall mass
**Treatment:** The patient was admitted to the hospital for preoperative preparation. There were no contraindications to surgery, so a radical gastrectomy was performed. No liver or pelvic metastases were found during the operation, there was no ascites, and the mass in the antrum of the stomach was about 2 cm in diameter, while the subserosal mass in the body of the stomach was 0.5 cm × 1.3 cm. The relevant lymph nodes were removed. After the operation, the patient was given anti-infective, anti-acid and rehydration treatments.

**o1**

**诊断 (score 4/5)：** 胃窦部印戒细胞癌
**依据：** (1) 临床症状：患者长期胃部不适，反酸、嗳气，进食后症状加重，偶有烧心，这些症状提示胃部可能存在病变。
(2) 影像学检查：CT显示胃壁局部增厚，提示胃壁有异常增生或肿瘤可能。
(3) 病理检查：胃窦部活检组织病理显示印戒细胞癌，确诊为胃癌的一种特殊类型。
**治疗(score 4/5)：** (1) 手术治疗：建议尽快进行根治性手术切除，包括患病部位的胃切除及必要的淋巴结清扫，以去除原发肿瘤和可能的转移病灶。具体手术方式需由胃肠外科医生根据实际情况确定。
(2) 化疗：根据肿瘤分期和手术情况，可能需要术前新辅助化疗或术后辅助化疗，以缩小肿瘤、消灭残留癌细胞，降低复发风险。
(3) 营养支持：术前术后应加强营养支持，保持良好的营养状态，有助于提高治疗耐受性和促进康复。
(4) 心理支持：确诊癌症可能带来心理压力，建议寻求专业的心理支持，保持积极乐观的心态。
(5) 定期复查：治疗后需要定期随访，进行胃镜、影像学等检查，监测病情变化，及时发现和处理可能的复发或转移。

**Diagnosis(score 4/5):** cancer of the gastric antrum with signet ring cells **(score 4/5).**
**Basis:** (1) Clinical symptoms: The patient had long-term stomach discomfort, acid reflux, belching, and occasional heartburn, which worsened after eating. These symptoms suggested that there might be a lesion in the stomach.
(2) Imaging examination: CT showed local thickening of the stomach wall, suggesting abnormal growth or the possibility of a tumor.
(3) Pathology: Biopsy of the antrum of the stomach showed signet ring cell carcinoma, a specific type of gastric cancer.
**Treatment(score 4/5):** (1) Surgical treatment: Radical surgical resection, including gastrectomy of the diseased area and necessary lymph node dissection, is recommended as soon as possible to remove the primary tumor and possible metastatic lesions. The specific surgical method needs to be determined by the gastrointestinal surgeon based on the actual situation.
(2) Chemotherapy: Depending on the stage of the tumor and the surgical situation, preoperative neoadjuvant chemotherapy or postoperative adjuvant chemotherapy may be required to reduce the tumor, eliminate residual cancer cells, and reduce the risk of recurrence.
(3) Nutritional support: Preoperative and postoperative nutritional support should be strengthened to maintain a good nutritional status, which helps improve treatment tolerance and promote recovery.
(4) Psychological support: A cancer diagnosis can bring psychological stress. It is recommended to seek professional psychological support to maintain a positive and optimistic attitude.
(5) Regular follow-ups: After treatment, regular follow-ups are required to monitor changes in the condition and to detect and deal with possible relapses or metastases in a timely manner.

**GPT-4**

**诊断(score 1/5)：** 胃食管反流病（GERD）
**依据：** (1) 症状描述，包括胃部不适、气体冒上来和酸水反流等典型胃食管反流病表现
(2) 病史中有胃溃疡，可能增加胃食管反流病的风险
(3) 上消化道内镜检查排除了胃黏膜损伤或活跃性溃疡等其他可能性
**治疗(score 1/5)：** (1) 饮食调：避免油腻、辛辣等刺激性食品，少量多餐，避免过饱
(2) 药物治疗：咨询医生考虑使用质子泵抑制剂（PPI）或组胺受体拮抗剂，以减少胃酸分泌和缓解症状
(3) 生活方式改善：保持健康体重，避免躺下后立即饮食，并停用或减少可能加剧症状的药物（需在医生指导下进行）

**Diagnosis(score 1/5):** Gastroesophageal reflux disease (GERD)
**Basis:** (1) Description of symptoms, including typical GERD symptoms such as stomach discomfort, gas coming up, and acid reflux (2) A history of a stomach ulcer, which may increase the risk of GERD (3) An upper endoscopy that ruled out other possibilities such as damage to the stomach lining or an active ulcer
**Treatment(score 1/5):** (1) Diet: Avoid greasy, spicy and other irritating foods, eat small amounts at frequent intervals, and avoid overeating (2) Medication: Consult your doctor about using proton pump inhibitors (PPI) or histamine receptor antagonists to reduce gastric acid secretion and relieve symptoms
(3) Lifestyle improvements: Maintain a healthy weight, avoid eating or drinking immediately after lying down, and stop or reduce the use of medications that may worsen symptoms (under the guidance of a doctor)

Figure 7: Comparison of the answers from GPT-o1 and GPT-4 for a case from the Chinese dataset AI Hospital, along with its English translation. o1 offers a more precise diagnosis and practical treatment suggestions compared to GPT-4.

