# OpenReview forum: "A Preliminary Study of o1 in Medicine: Are We Closer to an AI Doctor?"
_ICLR.cc/2025/Conference — ICLR 2025 Conference Withdrawn Submission_

### Official Review · Reviewer_w1Px · 2024-10-16

**Soundness:** 1
**Presentation:** 3
**Contribution:** 1
**Rating:** 3
**Confidence:** 5

**Summary:**

The work presented aims to evaluate the medical capabilities of OpenAI's latest `o1` model using 37 medical benchmarks, including tasks in concept recognition, clinical decision support and knowledge QA. The authors use a different set of metrics depending on the benchmarks, such as accuracy, f-score, BLEU/ROUGE and AlignScore. The authors compare the results with previous models `GPT-4`, `GPT-3.5` and open-source models `Meditron-70b` and `Llama3 8b`. The results demonstrate overall higher raw scores compared to previous models. Additional results demonstrate that using CoT prompting techniques on the older models improve their performance. The authors briefly discuss the limitations of the metrics used and the need for new evaluation methods. Based on these results, the authors conclude that these scores demonstrate that the `o1` model is closing the gap with human physicians.

**Strengths:**

The paper addresses a timely and important topic: the evaluation of the medical capabilities of OpenAI's latest o1 model. This focus is particularly relevant given the increasing deployment of these models in clinical settings. The paper is generally well-written and structured, which aids in navigating its content.

The paper uses a wide range of benchmarks, uses different evaluation metrics and introduces two new benchmarks which contributes to demonstrating that the improvements of o1 over previous models is not limited to a few benchmarks.

The analysis in section 4.3 and the discussion provide valuable insights. These sections accurately describe the limitations of the metrics used and highlight the need for new evaluation methodologies that more accurately assess medical capabilities. This critical perspective on current evaluation methods is the paper's most significant contribution.

The appendix contains useful additional data, including the prompts used and decoding times, which adds a degree of transparency to the study. The inclusion of case studies, while limited, provides concrete examples that complement the main text.

**Weaknesses:**

While the paper addresses a gap in literature, the methodology used, the reported results and especially their interpretation raise major concerns.

## Evaluation pipeline

1. The use of BLEU and ROUGE metrics for assessing tasks such as summarization is questionable, given that these metrics primarily measure n-gram overlap. This concern is further amplified by the results in Tables 3 and 4, which demonstrate no correlation between AlignScore and the BLEU/ROUGE results. This discrepancy raises doubts about the interpretation of results for these tasks.

2. The model selection appears limited in scope. A more comprehensive comparison would include comparable models from competitors, such as Claude 3.5 Sonnet. Additionally, the decision to compare OpenAI's latest and most powerful model to the 8b parameter version of Llama 3 is questionable. A more appropriate comparison would involve evaluating Llama 3 with 405b parameters.

## Experiments

3. There are significant inconsistencies in the reported results, raising concerns about their validity. For instance:
    - GPT-4 and GPT-3.5 are reported to achieve 52.8% and 25.4% on PubMedQA, respectively. However, previously reported results by Microsoft and OpenAI were 75.2% and 71.6% in zero-shot settings [1].
    - Inconsistencies exist between tables, such as the apparent inversion of MedQA and MedMCQA results in Table 6 compared to Table 2.
    - The NEJMQA results are not whole numbers, which is puzzling given that only 100 questions were used for evaluation.
The lack of reproducible code exacerbates these concerns.

4. The results presented in Table 2 are incomplete, with Meditron and Llama 3 lacking results for MedBullets, LancetQA, NEJMQA, and MedCalc-Bench.

5. The reported results, while adequately formatted, lack statistical significance or confidence intervals. This omission is particularly problematic for benchmarks that were subsampled due to budget constraints, such as NEJMQA (evaluated with only 100 questions). A t-test comparing the results of o1 and GPT-4 on this benchmark yields a p-value > 0.05, indicating no statistically significant difference.

6. The main conclusion (Section 4.2) stating "Yes! WE ARE ONE STEP CLOSER TO AN AI DOCTOR" is not sufficiently supported by the evidence presented. The paper does not demonstrate that higher performance on these benchmarks correlates with medical proficiency. Modern models of clinical reasoning emphasize that medical decision-making does not rely solely on internal cognition [2]. Even with proper evaluations of internal cognitive processes, additional assessments addressing the various cognitive processes involved in clinical reasoning would be necessary to support such a conclusion.

## Case study

7. The case study presented in Figure 6 is incomplete and potentially misleading. The NEJM question referenced was originally an image challenge [3]. It appears that the models were not provided with the necessary images, as evidenced by GPT-4's response mentioning "radiographic findings without seeing the images." This suggests that 'o1' may have hallucinated the radiographic findings ("isolated ulnar fracture") and arrived at the correct answer for incorrect reasons.

## References

[1] Nori, H., King, N., McKinney, S.M., Carignan, D., & Horvitz, E. (2023). Capabilities of GPT-4 on Medical Challenge Problems. ArXiv, abs/2303.13375.

[2] Parsons, A. S., Wijesekera, T. P., Olson, A. P. J., Torre, D., Durning, S. J., & Daniel, M. (2024). Beyond thinking fast and slow: Implications of a transtheoretical model of clinical reasoning and error on teaching, assessment, and research. Medical Teacher, 1–12. https://doi.org/10.1080/0142159X.2024.2359963

[3] NEJM Challenge 2023/04/20, https://www.nejm.org/image-challenge?ci=20230420

**Questions:**

While the paper in its current form is not suitable for publication, it contains potentially valuable insights that could significantly contribute to the field. The most interesting finding appears to be the inadequacy of current testing methodologies, particularly ROUGE/BLEU, in evaluating free-form text medical capabilities. For critical domains such as medicine, it is paramount that the BioNLP community uses appropriate methodologies to ensure patient safety.

## Reframing the paper

The paper could be substantially improved by reframing its focus on the limitations of current free-form text evaluations in medical AI. This reframing should include a comprehensive review of existing evaluation methods, a detailed analysis of why ROUGE/BLEU metrics are insufficient for medical text evaluation, and proposals for more appropriate evaluation techniques. By shifting the focus, the paper could make a significant contribution to the ongoing discussion about responsible AI development in healthcare.

## Methodology

1. The statistical rigor needs enhancement. This includes conducting and reporting appropriate statistical analyses to compare model results, including confidence intervals or p-values for reported metrics.

2. The benchmark selection and reporting require justification and improvement. The inclusion of benchmarks with small sample sizes, such as NEJMQA with only 100 samples, raises concerns about result reliability. The authors should either exclude these or discuss their limitations extensively.

3. For custom benchmarks like LancetQA and NEJMQA, a detailed methodology is needed. This should explain how questions with images or non-textual media were handled, the process for removing or substituting images with textual findings, and the potential impacts of these modifications on benchmark validity.

4. Incorporating human evaluators for at least a portion of the free-form text outputs would significantly enhance the paper's credibility. This should include a description of the human evaluation methodology, rater qualifications, and inter-rater reliability measures.

5. The results could be strengthened by evaluating more appropriate models such as Claude 3.5 Sonnet and Llama3.1 405b in addition to the existing evaluations.

6. A thorough review of the benchmarking code is necessary to address discrepancies with previous work. Making the evaluation code publicly available would allow independent verification of results, enhancing transparency and reproducibility.

## Results

7. The authors should revise their interpretation of results, avoiding direct links between BLEU/ROUGE metrics and "real-world clinical understanding" without supporting evidence. A more nuanced discussion of what the results might indicate, acknowledging the limitations of the metrics used, would be more appropriate.

**Details Of Ethics Concerns:**

The paper's methodological flaws and overreaching conclusions raise concerns, but it's unclear if they warrant a formal ethics review. While primarily scientific issues, there's a potential risk of indirect harm from overconfidence in the evaluated AI system. The paper highlights the need for more rigorous evaluation in medical AI but falls in a gray area regarding ethical implications. Standard peer review may suffice, but the decision is not straightforward.

---

### Official Review · Reviewer_jDqo · 2024-10-24

**Soundness:** 2
**Presentation:** 2
**Contribution:** 2
**Rating:** 3
**Confidence:** 4

**Summary:**

The paper evaluates OpenAI's latest large language model, "o1," in the context of medical applications. Focusing on understanding, reasoning, and multilingual capabilities, the authors assess o1's performance across various medical scenarios using six tasks derived from 37 medical datasets. Notably, they introduce two new and more challenging QA tasks—LancetQA and NEJMQA—based on professional medical quizzes from The Lancet and the New England Journal of Medicine. The findings suggest that o1 surpasses the previous GPT-4 model by an average of 6.2% and 6.6% in accuracy across the evaluated datasets. Despite these improvements, the paper also highlights weaknesses such as hallucinations, inconsistent multilingual abilities, and discrepancies in evaluation metrics.

**Strengths:**

- Comprehensive Evaluation: The paper attempts a broad assessment of o1's capabilities in the medical domain, covering understanding, reasoning, and multilinguality.

-  Introduction of New Datasets: By creating LancetQA and NEJMQA, the authors provide more clinically relevant and challenging benchmarks that may better reflect real-world medical scenarios.

**Weaknesses:**

- Datasets like MedQA, MedMCQA, and PubMedQA are categorized under "Reasoning" despite being primarily assessments of medical knowledge. This raises concerns about whether the tasks truly evaluate reasoning or merely recall of information. The methodology for evaluating reasoning appears limited, relying on static questions and accuracy metrics rather than assessing the reasoning process.

- Results for models like Meditron and Llama are taken from other papers, leading to discrepancies in evaluation methods. For instance, the significant performance difference on the PMC-Patient dataset suggests that evaluation schemas may not be consistent. The paper also does not include experiments with competitive models like Claude or Gemini, limiting the robustness of the comparative analysis.

- The creation and justification for LancetQA and NEJMQA are inadequately described. The paper lacks details on how these datasets were constructed, their content, and why they were necessary.

- Statements about advanced prompting techniques affecting o1's performance are based on limited data from a single dataset (LancetQA). This is insufficient to generalize conclusions about the efficacy of such techniques.

- Overall, the experiments and explanations are too insufficient concerning the various points claimed in the paper. It feels like this paper is
 just simply testing GPT o1 on several benchmarks and reporting the scores.

**Questions:**

Major Comments

1. In line 142, the authors state: "Our experimental findings reveal that with enhanced understanding, reasoning, and multilingual medical capabilities, o1 makes a step closer to a reliable clinical AI system."
What specific experiments demonstrate that o1 has moved closer to a reliable clinical AI system due to these "enhanced understanding, reasoning, and multilingual medical capabilities"?

2. Related to the first point, in line 151, the authors mention: "understanding, reasoning, and multilinguality, that correspond to the real-world needs of clinical physicians."
This claim requires supporting evidence. How does multilinguality specifically relate to the real-world needs of clinical physicians?

3. I am curious why MedQA, MedMCQA, and PubMedQA are categorized under "Reasoning" rather than "Understanding." According to the paper (line 226), "Understand refers to the model’s ability to utilize its internal medical knowledge," and in line 183, MedQA is described as "QA data for medical knowledge assessment."
Are all datasets classified under KnowledgeQA tasks truly indicators of reasoning ability? In the paper "Diagnostic reasoning prompts reveal the potential for large language model interpretability in medicine," LLMs' reasoning capabilities are assessed by having physicians evaluate the reasoning process beyond just the answers. Simply measuring accuracy on static questions seems more related to assessing internal knowledge rather than reasoning. Could the authors clarify this categorization?

4. In Tables 2 and 3, the authors compare Meditron and Llama using results from other papers rather than conducting direct experiments.
Why didn't the authors directly experiment with Llama and Meditron? For instance, on the PMC-Patient dataset, o1 scores 76.4 while Llama3-8b scores 96.0, indicating a mismatch in evaluation schemes. The current comparisons mainly involve o1, GPT-4, and GPT-3.5, effectively excluding Llama and Meditron. The comparison models seem insufficiently robust. Would it not strengthen the study to include models like Claude or Gemini?

5. The authors introduce LancetQA and NEJMQA datasets but provide limited details about them.
Why were LancetQA and NEJMQA created, and how were they constructed? The paper lacks detailed explanations beyond a brief mention in the appendix (line 967): "LancetQA and NEJMQA are datasets curated from The Lancet and the New England Journal of Medicine case challenges, focusing on patient diagnosis based on symptoms. We use 200 samples for LancetQA and 100 samples for NEJMQA." More comprehensive information about the purpose and construction of these datasets is necessary.

Minor Comments

6. The Abstract and Introduction mention benchmarking on 37 medical datasets.
Why does Figure 2 report averages based on only 19 datasets?

7. At line 162, it's stated: "Asterisks (*) denote the newly constructed datasets from public sources."
There are no asterisks in the table.

8. The experiments include Meditron-70b but not Llama3-70b.
Why was Llama3-70b excluded from the experiments? Considering that Meditron-70b is based on further medical pretraining of Llama2-70b, comparing Meditron-70b with Llama2-70b could offer valuable insights.

9. The authors claim (line 303): "We investigate the effect of several advanced promptings in our evaluation," and (line 431): "When it comes to other fancy promptings, such as self-consistency and reflexion, this conclusion may not stand."
Is it sufficient to make these conclusions based on a single dataset (LancetQA)? It seems additional data would strengthen these claims.

**Details Of Ethics Concerns:**

No ethics concerned in this work.

---

### Official Review · Reviewer_dg45 · 2024-11-04

**Soundness:** 3
**Presentation:** 4
**Contribution:** 2
**Rating:** 5
**Confidence:** 4

**Summary:**

This paper represents the first systematic evaluation of o1 medical capabilities within the community, providing a reference framework for the medical LLM community through comprehensive dataset collection. Through thoughtful analysis of the experimental process and results, the authors identify existing shortcomings in o1 and raise practical issues, such as the lack of evaluation metrics, for the community's consideration.

**Strengths:**

- This is the first systematic evaluation of o1 medical capabilities, providing the community with a glimpse into o1's medical reasoning abilities and offering first-hand data insights.

- Provide a detailed analysis of o1's experimental results, highlighting the inconsistency in performance rankings across different computational metrics for open-ended question-answering tasks.

- The authors found that o1 can still benefit from Chain-of-Thought (CoT) techniques and remains constrained by the issue of hallucinations.

- Introduce two novel and challenging QA datasets.

**Weaknesses:**

> Limited Contribution to the Community in Terms of Data and Technology:

- In terms of data, the article introduces two novel and challenging datasets, but it does not clearly state whether they will be open-sourced. Additionally, the processes of data scraping, processing, proofreading, and the total number of items for both datasets are not described in the main text. For other collected datasets, the article does not design cross-experiments or conduct secondary processing, resulting in limited insights derived from data collection and experimental evaluation design, aside from those provided by the o1 model itself.

- Technologically, the paper raises issues related to text similarity computation metrics, prompt design, and hallucination through experimental analysis, yet it does not attempt to mitigate even one of these problems.

> Minor Flaws in Presentation and Result Analysis:

The results presented in the paper, particularly the surprising findings that "o1 still benefits from CoT" and "o1's hallucination phenomenon remains serious," lack examples to help readers grasp these concepts intuitively. Furthermore, the authors should provide more analysis, possibly incorporating examples or statistical data, to explain the reasons behind these phenomena.

**Questions:**

Q1: Can the authors provide an accurate and detailed description of the data scraping, processing, proofreading, total number of items, and whether the two datasets will be open-sourced?

Q2: Are there any additional examples or detailed explanations that could further explore the reasons and extent of the phenomena "o1 still benefits from CoT" and "o1's hallucination phenomenon remains serious"?

Q3: Regarding the hallucination issue with o1, can it be improved through prompt or encoding parameters adjustments? Does this issue occur randomly, or is it consistently present in certain types of questions?

---

### Note · Authors · 2024-11-21

**Comment:**

I have read and agree with the venue's withdrawal policy on behalf of myself and my co-authors.

**Withdrawal Confirmation:**

I have read and agree with the venue's withdrawal policy on behalf of myself and my co-authors.